# Characteristics of ABCC4 and ABCG2 High Expression Subpopulations in CRC—A New Opportunity to Predict Therapy Response

**DOI:** 10.3390/cancers15235623

**Published:** 2023-11-28

**Authors:** Jakub Kryczka, Joanna Boncela

**Affiliations:** Laboratory of Cell Signaling, Institute of Medical Biology, Polish Academy of Sciences, 93-232 Lodz, Poland; jkryczka@cbm.pan.pl

**Keywords:** ABCC4, ABCG2, CRC, immune cell infiltration, metastasis, CRC subpopulations, CRC diagnostic and prognostic biomarkers

## Abstract

**Simple Summary:**

Colorectal cancer (CRC) is one of the most common malignancies worldwide, causing thousands to die each year. Its complex molecular nature leads to significant heterogeneity and variable responses to therapy. ABC proteins, which for many years were regarded as the pillar of the resistance to chemotherapy because they export anticancer drugs from cancer cells, have recently been identified as interesting molecular markers associated with many other physiological functions. We previously reported that during the phenotypic transition, CRC differentially regulates the expression of two transporters, ABCC4 and ABCG2. In cells with a mesenchymal and invasive phenotype, ABCC4 is upregulated, and ABCG2 is downregulated. We have therefore decided to explore this phenomenon by analysing samples from CRC patients with high expression of either ABCC4 or ABCG2 to determine their potential use as markers of therapeutic outcome.

**Abstract:**

Background: Our previous findings proved that ABCC4 and ABCG2 proteins present much more complex roles in colorectal cancer (CRC) than typically cancer-associated functions as drug exporters. Our objective was to evaluate their predictive/diagnostic potential. Methods: CRC patients’ transcriptomic data from the Gene Expression Omnibus database (GSE18105, GSE21510 and GSE41568) were discriminated into two subpopulations presenting either high expression levels of ABCC4 (ABCC4 High) or ABCG2 (ABCG2 High). Subpopulations were analysed using various bioinformatical tools and platforms (KEEG, Gene Ontology, FunRich v3.1.3, TIMER2.0 and STRING 12.0). Results: The analysed subpopulations present different gene expression patterns. The protein–protein interaction network of subpopulation-specific genes revealed the top hub proteins in ABCC4 High: RPS27A, SRSF1, DDX3X, BPTF, RBBP7, POLR1B, HNRNPA2B1, PSMD14, NOP58 and EIF2S3 and in ABCG2 High: MAPK3, HIST2H2BE, LMNA, HIST1H2BD, HIST1H2BK, HIST1H2AC, FYN, TLR4, FLNA and HIST1H2AJ. Additionally, our multi-omics analysis proved that the ABCC4 expression correlates with substantially increased tumour-associated macrophage infiltration and sensitivity to FOLFOX treatment. Conclusions: ABCC4 and ABCG2 may be used to distinguish CRC subpopulations that present different molecular and physiological functions. The ABCC4 High subpopulation demonstrates significant EMT reprogramming, RNA metabolism and high response to DNA damage stimuli. The ABCG2 High subpopulation may resist the anti-EGFR therapy, presenting higher proteolytical activity.

## 1. Introduction

Colorectal cancer (CRC) remains one of the most common cancers and the leading cause of cancer-related mortality worldwide [1]. Currently, the most effective treatment for CRC is primary tumour resection with adequate histologic margin, often preceded or followed by adjuvant chemotherapy [2]. However, approximately 25% of patients with CRC will develop distant metastases at the time of initial diagnosis, which is the leading cause of cancer-related mortality. Additionally, up to 50% of patients develop distant metastases as the disease progresses. Predominant sites of CRC metastasis are the liver, lung and peritoneum [3]. CRC is a highly heterogeneous cancer. This results from the cellular plasticity of epithelial-to-mesenchymal transition (EMT) and the different sites of origin. Proximal colon (right-sided) tumours predominantly show flat histology and mutations in the DNA mismatch repair pathway. In contrast, distal colon (left-sided) tumours show polypoid morphology and mutations related to the chromosome instability pathway, such as KRAS, APC, PIK3CA and p53 [4]. The heterogeneity of CRC is a key determinant of their variable response or resistance to therapy. Unfortunately, CRC is one of the most therapy-resistant malignancies, highly unresponsive to immunotherapy and various chemotherapeutic regimens based on a combination of 5-fluorouracil (5FU), oxaliplatin (OxP) and irinotecan (IRI) [2,5,6,7]. The activity of specific transporters belonging to the ATP-binding cassette (ABC) protein family, such as ABCB1, the ABCC family, and ABCG2, has been closely associated with both acquired and innate chemoresistance due to their ability to export large amounts of various xenobiotics [8,9]. In CRC, the clinical studies focused mainly on ABCG2 and its role in irinotecan response. ABCG2 mRNA expression was found to be lower in tumours than in normal colonic tissue. These data suggest that primary colon cancer cells initially downregulate ABCG2 mRNA expression [10]. Our previous studies show that CRC overexpressing Snail, an EMT-initiating transcription factor, reveals upregulation in ABCC4 and downregulation in ABCG2 protein expression [11,12]. These results may indicate that ABCC4 expression is associated with the acquisition of mesenchymal features by cells, and, in a more general sense, the expression pattern of ABCC4/ABCG2 may be a determinant of phenotypic transition in CRC. We analysed microarray data from the public Gene Expression Omnibus (GEO) database to confirm this observation. We found that ABCC4 was significantly upregulated, whereas ABCG2 was downregulated in primary tumours in comparison to normal colon tissue.

Interestingly, ABC expression profiles constantly change during ongoing EMT and cancer progression. In recent years, an increasing number of studies have shown that loss or inhibition of ABC transporters affects cellular phenotypes closely associated with differentiation, migration/invasion and malignant potential in various cancers. In addition, loss of ABC transporters in both xenograft and transgenic mouse models of cancer can affect tumour initiation and progression. [13,14,15]. These effects are probably a result of their normal physiological function as exporters of endogenous metabolites and signalling molecules. Thus, ABC transporters play a much more complex role in cancer development than drug efflux [11,16]. To further investigate the importance of ABC proteins and their engagement in different cancer-related processes, in this manuscript, we decided to compare two CRC subgroups presenting high expression levels of two ABC members: ABCC4 and ABCG2.

## 2. Material and Methods

### 2.1. Microarray Data Processing and Analysis

Gene expression profiles with accession numbers GSE18105 (https://www.ncbi.nlm.nih.gov/geo/query/acc.cgi?acc=GSE18105, accessed on 4 September 2023), GSE21510 (https://www.ncbi.nlm.nih.gov/geo/query/acc.cgi?acc=GSE21510, accessed on 4 September 2023) GSE41568 (https://www.ncbi.nlm.nih.gov/geo/query/acc.cgi?acc=GSE41568, accessed on 4 September 2023), GSE83129 (https://www.ncbi.nlm.nih.gov/geo/query/acc.cgi?acc=GSE83129, accessed on 4 September 2023) and GSE62080 (https://www.ncbi.nlm.nih.gov/geo/query/acc.cgi?acc=GSE62080, accessed on 4 September 2023) were downloaded from The Gene Expression Omnibus (GEO) database (http://www.ncbi.nlm.nih.gov/geo/) (accessed on 4 September 2023) and analysed similarly to our previous work [16]. All data were processed using the GEO2R online analytical tool (# Version info: R 4.2.2, Biobase 2.58.0, GEOquery 2.66.0, limma 3.54.0) [17]. Linear projections of gene mRNA level were performed using Orange 3.31.1 software as previously presented by us [16]. mRNA levels were calculated and visualised using JASP 0.16.0.0 software (https://jasp-stats.org/, accessed on 4 September 2023), as shown in [18].

### 2.2. Survival Probability Analysis

The survival rate for patients presenting the high and low expression of chosen differently expressed genes (DEGs) was analysed in CRC patients using TCGA data, the Human Protein Atlas (www.proteinatlas.org, accessed on 4 September 2023) “pathology” section [19,20] and TIMER2.0 platform (http://timer.cistrome.org/, accessed on 4 September 2023) [21]. The presented data used the best expression cut-off suggested by HPA.

### 2.3. Enrichment Analysis

Functional enrichment software tool FunRich (v3.1.3) (http://www.funrich.org/, accessed on 4 September 2023) supported by the Gene Ontology (GO) (http://geneontology.org/, accessed on 4 September 2023) database was used to compare, analyse and visualise the Biological Process (BP) and Molecular Functions (MF) differences associated to the proteins encoded by differently expressed genes (DEGs) in ABCC4 High- and ABCG2 High-level presenting subfractions of CRC patient samples analogues to our previous work [22].

### 2.4. Hierarchical Clustering

The top proteins upregulated in both the ABCC4 High and ABCG2 High CRC subgroups were used to create a bidirectional hierarchical clustering heatmap with their respective mRNA levels. The hierarchical clustering method results in a hierarchical dendrogram highlighting similarities and differences between the subjects analysed. The calculation and visualisation were performed using the Orange open source machine learning and data visualisation platform 3.31.1 (https://orangedatamining.com/, accessed on 4 September 2023), as previously described [16,22].

### 2.5. Protein–Protein Interaction Network

Protein–protein interaction (PPI) networks of top proteins expressed by ABCC4 High and ABCG2 High CRC subgroups were created and visualised using the STRING version 12.0 online platform (https://string-db.org/, accessed on 4 September 2023) and Cytoscape 3.9.1. as presented by us [16,22,23].

### 2.6. Analysis of Immune Cell Tumour Infiltration

The Tumor Immune Estimation Resource—TIMER2.0 platform (http://timer.cistrome.org/, accessed on 4 September 2023) was used to analyse immune cell infiltration. TIMER2.0 employs immunedeconv—an R package that integrates six state-of-the-art algorithms (TIMER, xCell, MCP-counter, CIBERSORT, EPIC and quanTIseq) to statistically predict tumour infiltration by selected immune cell types using The Cancer Genome Atlas (TCGA) database. Similar to our previous work, the data were analysed and visualised using the xCell algorithm [22].

### 2.7. Statistics

Statistical evaluation was performed using the normality test (Shapiro–Wilk), followed by the Student’s *t*-test (in the case of normally distributed data) or the Mann–Whitney U test (in the case of non-normally distributed data). Calculations and graphs were performed using Orange data mining 3.31.1 software and JASP 0.16.0.0 software; *p* values < 0.05 were considered statistically significant for all analyses: * *p* < 0.05; ** *p* < 0.005; *** *p* < 0.001, NS-not statistically significant. Pearson’s linear correlation analysis was performed using JASP 0.16.0.0 software with Pearson correlation coefficient presented as colour intensity and numerical values on the correlation matrix. The correlation statistical value was shown as follows: * *p* < 0.05; ** *p* < 0.005; *** *p* < 0.001, no indication—not statistically significant.

## 3. Results

### 3.1. Analysis of ABCC4 and ABCG2 Expression Level in CRC

Data containing mRNA levels of approximately 40,000 “hits” detected by microarrays chips in colorectal cancer tumours and normal colon tissue were downloaded from the GEO database (https://www.ncbi.nlm.nih.gov/geo/, accessed on 4 September 2023). Two datasets were analysed: GSE18105 (composed of n = 111) and GSE21510 (composed of n = 148). Expression of ABCC4 and ABCG2 in CRC samples and normal colon tissue was analysed in each dataset independently, as shown in Figure 1A–D. ABCC4 presents significantly higher expression in CRC samples than in normal colon tissue, whereas ABCG2 expression is considerably higher in normal colon tissue than in CRC. Interestingly, the TIMER2.0 platform [24]-based analysis of the TCGA database proves that neither ABCC4 (Figure 1E) nor ABCG2 (Figure 1F) expression level presents any statistically significant association with survival rate. Additionally, further analysis performed with the TIMER2.0 platform and TCGA database proved that expression of the mutated (including any type of mutation) variant of ABCC4 negatively correlates with the expression level of wild type (WT) of ABCC4 (Pearson correlation *p* = −0.233) and positively with ABCG2 (Pearson correlation efficiency *p* = 0.666) (Figure 1G,H).

### 3.2. Correlation of Immune Cell Infiltration with ABCG2 and ABCC4 Expression Levels in CRC

The tumour microenvironment (TME) is composed of cancer cells, normal cells (tissue of origin), cancer-associated fibroblasts (CAFs) and various immune cells such as CD8 + T cells, natural killer cells (NK cells), regulatory T cells (Treg cells), tumour-associated macrophages (TAM), and Dendritic cells (DC). The cellular components of TME regulate tumour survival and promote metastasis. In recent years, many studies have investigated the role of tumour infiltration by immune cells [25,26]. In the case of CRC, intratumoral infiltration by CD8+ and CD4+ T cells is concerned with a favourable prognostic factor increasing patients’ overall survival rate, whereas M2 tumour-associated macrophages (M2-TAMs) infiltration promotes cancer cell proliferation and increases metastatic potential [27,28,29]. Using the TIMER2.0 platform and gene-signature-based algorithm—xCELL, we analysed the correlation of ABCC4 and ABCG2 mRNA levels with immune cell infiltration (Table 1 and Appendix A) [24,30]. Our analysis proves that both ABCC4 and ABCG2 levels present a negative correlation with CD4+ and CD8+ T-cell infiltration but a positive correlation with CAFs infiltration. Interestingly, the mRNA level of mutated ABCC4 shows a high positive correlation with CD4+ (CD4+ Th2 log2FC = 0.565 (Figure 2A) and CD4+ Th1 log2FC = 0.805 (Figure 2B)) and CD8+ (CD8+ central memory log2FC = 1.401 (Figure 2C) and CD8+ naive log2FC = 0.526 (Figure 2D)) T-cell infiltration, with no impact on other immune cells.

### 3.3. Identification of ABCC4 and ABCG2 High Expression CRC Subsets

Even though ABCG2 presents significantly lower expression in CRC samples than in the normal colon, a small subfraction showing a high mRNA level is observed. Additionally, CRC cells present various ranges of ABCC4 expression. Thus, we decided to identify and analyse differences between CRC subfractions presenting high ABCC4 (ABCC4 H) and high ABCG2 (ABCG2 H) levels. First, we have selected two CRC subgroups for each of the analysed datasets using Orange data mining 3.31.1 software and a VizRank-based algorithm (“linear projection”). The first subgroup presented a high ABCC4 level and low ABCG2 level, whereas the second subgroup presented the opposite expression pattern, as shown in Figure 3A,B [31]. Next, gene expression patterns specific to each subgroup were compared using the online tool GEO2R with an adjustable *p* value < 0.05 for every dataset (Figure 3C). This analysis provided 867 upregulated genes for the ABCC4 High subgroup and 918 upregulated genes for the ABCG2 subgroup in the GSE18105 dataset and analogously 970 and 1275 for GSE21510 (Figure 3D). Finally, using the Venn Diagram, 704 genes significantly upregulated in the ABCC4 High CRC subgroup and 772 genes upregulated in ABCG2 HIGH CRC subgroups were identified in both datasets, as shown in Figure 3E.

### 3.4. Enrichment Analysis of DEGs Unique to ABCC4 High or ABCG2 High CRC Subsets

Genes upregulated in ABCC4 High and ABCG2 High CRC subgroups were analysed using the FunRich platform supported by the Gene Ontology database to verify differences in enrichment of Biological Processes (BP) (Figure 4A) and Molecular Functions (MF) (Figure 4B). The ABCC4 High CRC subgroup presents significantly higher enrichment in processes related to DNA and RNA binding, regulation of gene expression and response to DNA damage. In contrast, the ABCG2 High CRC subgroup demonstrates significant enrichment in positive regulation of apoptotic processes, cell adhesion, extracellular matrix decomposition, actin filament assembly and cell migration.

### 3.5. Correlation of ABCC4 and ABCG2 Expression Levels with Major Dysregulated Protein Hubs

Having established enriched biological processes, we shifted our focus to major dysregulated protein hubs. Thus, data containing upregulated genes from each subgroup were used to draw a protein–protein interaction (PPI) network via the STRING (ver. 12.0) platform (https://string-db.org/, accessed on 4 September 2023) (Appendix A). Next, the PPI network was analysed using Cytoscape 3.9.1 (https://cytoscape.org/, accessed on 4 September 2023) to identify the top 10 protein hubs (for each subgroup) with the highest number of direct protein interaction counted as the highest number of drawn “edges” (Table 2), similar to our previous study [16]. This type of analysis provides insight into critical proteins that, by direct interactions, influence various processes and thus potentially can be utilised as molecular targets for future therapies. To further analyse and verify the correctness of chosen protein hubs, a hierarchical clustering analysis of data consisting of mRNA levels of 20 chosen DEGs from all CRC patients (GSE18105 and GSE21510) was performed (Figure 5A). Step by step, this analysis connects most similar subjects, forming clusters (branches) until all clusters are defined. The obtained hierarchical dendrogram proves that protein hubs upregulated in ABCC4 High CRC subgroups cluster together with ABCC4 on one arm (branch). In contrast, protein hubs were observed for the ABCG2 High subgroup on the second arm, together with ABCG2. Additionally, using Orange data mining 3.31.1 software and the FreeViz tool, ABCC4 High and ABCG2 High CRC patients cluster differentiation, using top protein hubs, was visualised (Figure 5B). Next, the Pearson correlation matrix was created to analyse the mutual interaction of genes that encode the chosen top networking protein hub (Figure 5C). Interestingly, selected protein hubs demonstrate a substantial amount of interaction with each other, forming stable clusters presented in Figure 6. The cluster formed for the CRC subgroup characterised by high ABCC4 (Figure 6A) expression enriches biological processes such as GO:0003723—RNA binding and GO:0003676—Nucleic acid binding (according to the Gene Ontology database). Arguably, two of the most important proteins of this cluster are RPS27A and NOP58. RPS27 shows the highest number of edges, directly interacting with 10% of all proteins observed in this group (87 out of 867 proteins), whereas NOP58 presents the highest connectivity inside the cluster. Both proteins play important antiapoptotic roles [32,33]. On the other hand, the ABCG2 High cluster (Figure 6B) presented enrichment in GO:0046982—Protein heterodimerisation activity, GO:0046983—Protein dimerisation activity and GO:0005515—Protein binding.

Additionally, using the STRING platform, we have added known proteins that interact with selected clusters, thus filling the gaps to obtain major KEGG pathways, with PPI enrichment *p*-value for ABCC4 High and ABCG2 High respective subgroups: 1.77 × 10^−14^ and 4.28 × 10^−5^, respectively. The obtained data are shown in Table 3. Interestingly, the obtained protein hub cluster network for the ABCC4 High subgroup presents high involvement in proteasome functions and RNA polymerase functions, whereas the protein hub cluster network for the ABCG2 High subgroup—in MAPK signalling pathway, focal adhesion, PD-L1 expression and PD-1 checkpoint pathway, platelet activation, natural-killer-cell-mediated cytotoxicity, apoptosis and insulin signalling pathway.

### 3.6. Analysis of Potential CRC Metastatic Organotropism Biomarkers

Finally, we decided to verify whether mRNA levels of ABCC4, ABCG2 and genes encoding major protein hubs for their respective CRC subgroups (ABCC4 High and ABCG2 High) can be used to predict metastatic organotropism. Thus, we have downloaded and analysed transcriptomic data from the GSE41568 dataset (composed of n = 133), consisting of primary CRC samples and CRC metastases to the liver, lung and omentum [34]. Samples were divided into four subgroups, Primary, Lung Met., Liver Met. and Omentum Met., and analysed using ANCOVA with Post Hoc Test and Tukey correction. The ABCC4 mRNA level presents no statistically significant changes that could distinguish metastatic sites. However, ABCG2 and six other genes encoding protein hubs (FYN, FLNA, POLR1B, RBBP7, EIF2S3 and PSMD14) were found to be potentially valuable (Figure 7A). According to our analysis, upregulation of ABCG2, FLNA and FYN with simultaneous downregulation of RBBP7 is characteristic of CRC metastasis to the liver. Thus, we may assume that the ABCG2 High CRC subgroup prefers liver metastasis. Additionally, POLR1B and EIF2S3 expression downregulation compared to primary CRC and liver metastasis was observed for CRC samples resected from the lung. Furthermore, FYN upregulation is also significant for omentum, peritoneal and abdominal wall metastasis. This metastatic site lies near the primary CRC tumour (compared to distant metastasis to the lungs or liver). It requires a different form of cancer cell migration, preferring single-cell mesenchymal migration focusing on adhesion/deadhesion and active decomposition of extracellular matrix (ECM). The protein encoded by FYN is highly involved in hsa04510 (focal adhesion) and hsa04520 (adherent junctions), as well as hsa04611 (platelet activation). Thus, to further combine the obtained data, we have developed a metastasis site discriminative model using Orange data mining 3.31.1 software and a linear projection tool based on the VizRank algorithm (Figure 7B). This model places each data point on the visualisation matrix, simultaneously analysing each attribute value (gene expression). All attributes—chosen genes—are identified on a ring surrounding the visualisation space and equally separated from one another (in this instance, 45^0^ or 0.785 RAD); thus, the greater the attribute value, the closer the data point is drawn in 2D space [31]. Most primary CRC is located inside the triangle created by EIF2S3, RBBP7 and FYN and most Lung metastasis inside the triangle formed by ABCC4, PSMD14 and FLNA. The proposed method could potentially be beneficial in predicting (with some probability) metastatic progression after primary CRC biopsy. In addition, using the TIMER2.0 platform [24] and data from the TCGA database, we analysed the correlation between the expression levels of FYN, FLNA, POLR1B, RBBP7, EIF2S3 and PSMD14 and the survival of CRC patients (Figure 8). The average 5-year survival rate for CRC is 60%, a high expression of FLNA and POLR1B represents a significantly lower probability of the survival of CRC patients’, but only FLNA can be considered as a prognostic biomarker with HR = 1.25 and *p* = 0.025.

### 3.7. Analysis of ABCC4 and ABCG2 Potential Chemotherapy Response Predictive Capabilities

Finally, to evaluate differences between ABCC4 and ABCG2 mRNA expression and response to the main anti-CRC chemotherapy regimens, two datasets containing mRNA expression levels of CRC patients treated with FOLFIRI (GSE62080 consisting of n = 21; 12 resistant and 9 sensitive) and FOLFOX (GSE83129 consisting of n = 23; 12 resistant, 21 sensitive) were downloaded from the GEO database. Data were analysed using previously selected and set gates for CRC subpopulations with high expression of ABCC4 and ABCG2 using Orange data mining 3.31.1 software and a VizRank-based algorithm (“linear projection”) (Figure 9A,B). The ABCC4 High subgroup consists mainly of FOLFOX-sensitive samples and moderately of FOLFIRI-resistant samples, whereas the ABCG2 High subgroup is an even mix of samples with different chemotherapy responses. In addition, analysis of ABCC4 and ABCG2 mRNA expression levels revealed no statistically significant differences between FOLFIRI (Figure 9C)- and FOLFOX (Figure 9D)-resistant and -sensitive CRC patients.

## 4. Discussion

CRC, being one of the most common cancer types, contributes highly to cancer-related mortality. One of the most critical reasons is high molecular heterogeneity within CRC, resulting in substantial differences in treatment response to both chemo- and immune-therapy [4]. Regarding various immunotherapy schemes, CRC is considered one of the most resistant cancer types. Approximately 40–50% of CRC presents resistance to anti-epidermal growth factor receptor (anti-EGFR) therapy, and only 5–10% of metastatic CRC shows a positive response to anti-PD-1/PD-L1 (anti-programmed death-1/programmed cell death ligand 1) therapy [35,36]. Thus, standard first-line treatment for CRC includes conventional, reasonably inexpensive, tissue non-specific chemotherapy based on a mix of fluorouracil and oxaliplatin or irinotecan, often leading to acquisition resistance to its components by CRC cells [5]. Different mechanisms of chemoresistance, such as systems of DNA damage repairs or overexpression of antiapoptotic factors, have been identified. However, one of the most extensively studied mechanisms is the expression of anticancer drug exporters belonging to the ABC transporter family [11,23,37]. In the case of CRC, the significance of ABCG2 protein level as the patient’s potential predictive marker of resistance to irinotecan was examined. ABCG2 protein expression analysed by IHC showed that ABCG2-positive cells were mainly positioned in the cancer’s invasion front, and strong membranous staining was significantly correlated with a higher Dukes’ stage and distant metastases [10,38]. On the other hand, high ABCG2 expression does not contribute to higher patient mortality and negatively correlates with EMT advancement [11]. In addition, our analysis showed that ABCG2 expression did not correlate with response to FOLFOX or FOLFIRI treatment. There were no statistically significant changes in mRNA expression in the cohorts of responding and non-responding patients. In contrast, CRC patients with high ABCC4 expression and low ABCG2 expression may be considered sensitive to FOLFOX therapy and resistant to the FOLFIRI regimen. This is probably due to substrate specificity. Both proteins transport irinotecan, but neither transports oxaliplatin [39]. Recent studies prove that ABC transporters use mitochondrial-derived ATP, but not ATP from glycolysis, as the primary source of energy for drug efflux in chemo-resistant cancer cells. Importantly, often observed among various cancer types, a metabolic switch toward aerobic glycolysis (known as the Warburg effect) renders ABC proteins rather non-related to chemoresistance [37,40,41]. Thus, the correlation between the level of ABC proteins and their actual molecular function in cancers remains to be proven.

Our multi-omics analysis proves that CRC patients’ sample subfractions, expressing a high level of either ABCC4 or ABCG2 followed by a low mRNA level of the second corresponding ABC transporter, are presenting different molecular and physiological functions, even though both transporters present to some extent substrate homology including active transport of irinotecan and its active metabolite SN38 [39]. The ABCG2 High CRC subgroup presents significant enrichment in positive regulation of apoptotic processes, cell adhesion extracellular matrix decomposition, actin filament assembly and cell migration. On the other hand, dysregulated genes in the ABCC4 High CRC subgroup significantly enrich processes related to DNA and RNA binding, regulation of gene expression and response to DNA damage. This observation corresponds to our previous findings, in which we have proven that during phenotypical reprogramming such as EMT, mesenchymal phenotype presenting CRC cells upregulate ABCC4 expression, simultaneously downregulating ABCG2 expression [11]. Differences in the number and composition of EMT transcription factors binding sites for both ABC transporters may explain their various expression during EMT. *ABCC4* poses 11 E-Box sequences and 3 TWIST binding sites, whereas *ABCG2*—only 6 E-Box sequences and 1 TWIST binding site [42].

Similar CAFs, Neutrophils, CD8+ and CD4+ T-cell infiltration patterns may characterise both analysed CRC subfractions. However, ABCC4 expression positively correlates with M2 macrophage infiltration, whereas ABCG2 expression level does not impact this process. Polarised M2 macrophages increase stemness and metastatic ability of CRC cells by secreting TGF-β2 and chemokine C-X-C-Motif Ligand 12 (CXCL12), which activates the WNT/β-catenin pathway, thus promoting EMT [43]. Therefore, we suggested that due to their complex functions, ABC transporters could be considered to represent markers of specific molecular alterations or advanced reprogramming that accompany cancer progression and that more attention should be placed on the most dysregulated genes and their role in the regulation of particular processes in each analysed subpopulation.

Top networking (top hub) proteins encoded by upregulated genes in the ABCC4 High CRC subgroup are mainly involved in the regulation of nearly all stages of RNA metabolism (transcription, pre-mRNA splicing, RNA export and translation), chromatin metabolism, transcription factors binding and construction of small nucleolar ribonucleoproteins (snoRNPs) complex (composed of core RNP proteins such as NOP58, and rRNA such as SNORD27, which facilitates 2′-O-Me of A27 on 18S rRNA) [33,44,45]. Gene expression and protein levels of NOP58, BPTF and SRSF1, HNRNPA2B1 correlate with CRC progression, increased proliferation and metastasis, TNM staging, and poor prognosis of CRC patients. RNA-binding protein HNRNPA2B1 is involved in the transportation and posttranscriptional regulation of numerous cancer-progression-related micro RNA (miRNAs) and long noncoding RNA (lncRNA) through binding of the specific motifs GGAG/CCCU such as miR-934, Linc01232, miR100HG, H19 and RP11 [46,47]. Recently, it was proven that in the case of CRC, HNRNPA2B1 mediated miR-934 packaging into exosomes that macrophages take up, leading to their polarisation into TGF-β2-secreting M2 macrophages [47]. The protein encoded by the *SRSF1* gene promotes alternative splicing of BIM (also known as BCL2L11) and BIN1—isoforms that lack pro-apoptotic functions [48]. BPTF promotes CRC cell cycle progression, thus CRC proliferation and overall tumour progression by targeting Cell division cycle 25 A (Cdc25A) [49]. NOP58 is a ribonucleoprotein that is a central component for several box C/D small nucleolar RNAs (snoRNAs), such as U3, U8 and U14. Its overexpression is associated with poor cancer patients’ survival, as it may regulate cell cycle mitosis, mitotic G1/S phase, mitotic G2/M phase, Rb-1 pathway, M phase, IL-10 signalling, pathways via regulation of *TP53* and P53 activity [50]. Interestingly, lncRNA ZFAS1, one of the significant EMT inducers in CRC, promotes small nucleolar RNA-mediated 2′-O-methylation via NOP58 recruitment and is also responsible for CRC tumorigenesis and further progression via DDX21-POLR1B regulatory axis [33,51,52]. Retinoblastoma-binding protein 7 (RBBP7), an important component of chromatin metabolism-regulating complex, is overexpressed in various cancer types, enhancing cancer cell proliferation, invasion and stemness, increasing cyclin-dependent kinase 4 (CDK4) expression [44,53]. The impact of DDX3X on CRC progression is somewhat enigmatic due to its involvement in all stages of RNA metabolism, which impacts different signalling pathways [45]. On the one hand, it has been reported to act as a tumour suppressor. On the other hand, DDX3X has been shown to induce EMT and subsequent CRC proliferation and migration by stabilising the mRNA of the transcription factor GATA2 [54,55]. Interestingly, the blood mRNA level of EIF2S3 was found to be a discriminating marker of CRC [56]. In addition, the expression levels of other top network proteins, such as RPS27A and PSMD14, are usually significantly higher in CRC tumours than in tumour-adjacent tissue, but unlike PSMD14, which is associated with more aggressive cancers, RPS27A correlates with smaller tumours, lower T-stage and drastically reduced apoptosis rates. [32,57].

In the ABCG2 High CRC subgroup, most top hub proteins belong to the histone cluster family. Recent studies proved that mRNA expression of HIST1H2BK, HIST1H2AG, HIST2H2AA4, HIST1H2BJ, HIST2H2BE and HIST1H2AC proteins positively correlates with each other, and their upregulation is related to the poor prognosis in glioma [58]. Additionally, HIST1H2BK correlates with metastatic CRC cytokine secretion, myeloid leukocyte migration into the tumour, and resistance to proteasome-inhibitor-based anticancer therapy [59].

Although a great deal of CRC patients present resistance to anti-EGFR therapy, it remains the primary form of immunotherapy administered to the patients [35,36]. Three (FLNA, TLR4 and FYN) of the top hub proteins for the ABCG2 High CRC subgroup are involved in the EGF–EGFR pathway. CRC tumours, characterised by high filamin A (FLNA) expression, do not respond to anti-EGFR therapy in cetuximab treatment but may respond to c-MET receptor tyrosine kinase inhibitors [60]. In squamous cell carcinoma, activation of TLR4 reversed cetuximab-induced inhibition of proliferation, migration and invasion, increasing resistance to anti-EGFR therapy [61]. As an effector of oncogenic EGFR signalling, Fyn promotes tumour growth and motility. Its silencing limits EGF-triggered cancer progression [62]. Interestingly, several multi-omics analyses prove that MAPK3 occupies one of the top networking protein positions along with EGFR in various cancers, resulting in poor prognosis [63,64].

From a clinical applicability perspective, precise discrimination of CRC patients into ABCC4/ABCG2 High subgroups may be another factor enabling the selection of correct and appropriate personalised therapy, including a mix of chemo and immunotherapy and identifying the most likely metastatic site.

## 5. Conclusions

Our analysis proved that ABCC4 and ABCG2 mRNA levels may be used to distinguish two molecular and physiologically different CRC subgroups that may present different susceptibilities to specific therapy. The CRC subgroup characterised by high expression of ABCC4 shows substantial dependence on EMT reprogramming (acquired via TMEM interaction) and RNA metabolism, with higher response to DNA damage stimuli and rather good response to oxaliplatin-based FOLFOX treatment that primarily focuses on the formation of DNA-platinum adducts. It may also be regulated by lncRNA ZFAS1, whereas ABCG2 high expression presenting CRC subgroup may be resistant to the anti-EGFR therapy, demonstrating higher proteolytical activity and actin-filament-related activity, thus higher ability to invade surrounding tissue. Unfortunately, the precise correlation of ABCC4 and ABCG2 mRNA expression levels with response to chemotherapy is limited by the small sample size (n) of FOLFOX- and FOLFIRI-treated patients. In addition, most of the data were obtained from Japanese (Asian) and Danish (Caucasian) patients, and accurate extrapolation to other ethnic groups is difficult and requires further investigation.

## Figures and Tables

**Figure 1 cancers-15-05623-f001:**
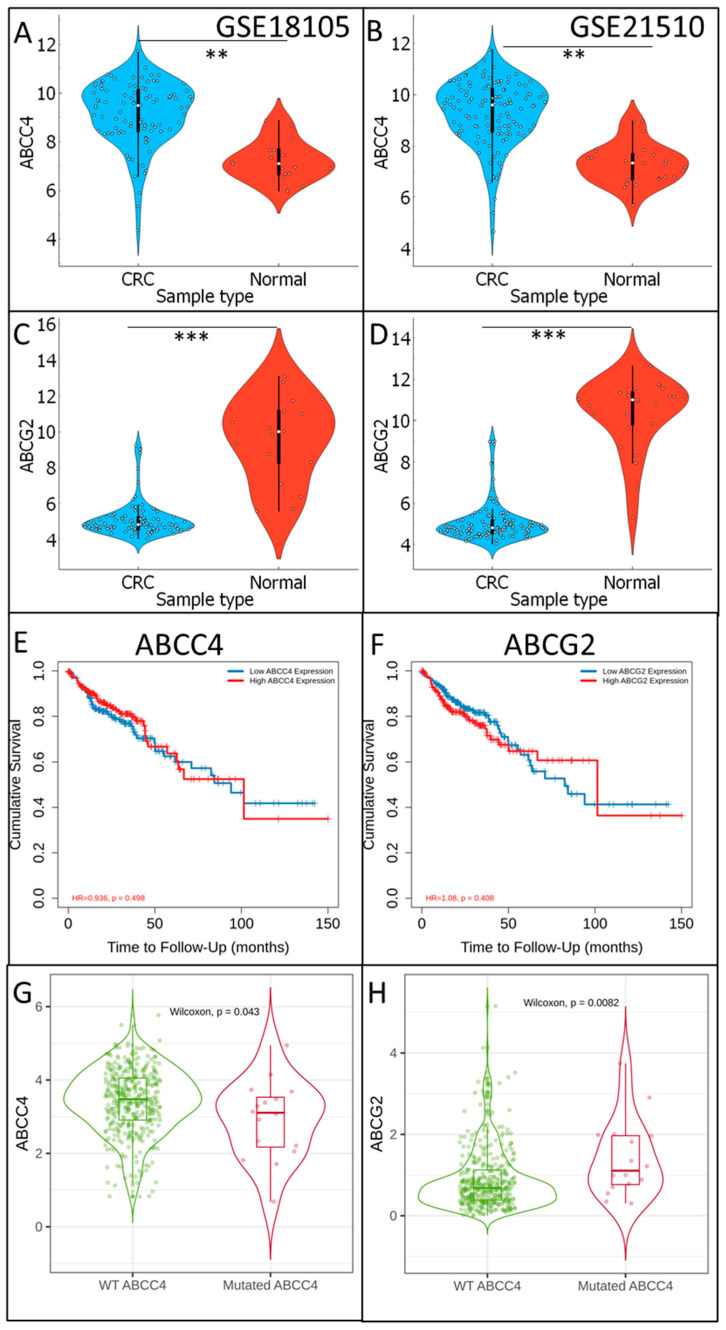
ABCC4 and ABCG2 involvement in CRC progression. ABCC4 and ABCG2 expression levels in “CRC” and noncancerous “Normal” colon tissue were calculated using data from GSE18105 (**A**,**C**) and GSE21510 (**B**,**D**) and visualised using Orange data mining 3.31.1 software. A normality test (Shapiro–Wilk) was performed, followed by the Mann–Whitney U test ** *p* < 0.005; *** *p* < 0.001; ABCC4 (**E**) and ABCG2 (**F**) impact on survival rate was analysed using TCGA data and visualised by the TIMER2.0 platform. The correlation of ABCC4 (**G**) and ABCG2 (**H**) wild-type (WT) and mutated variants was calculated using TCGA data and visualised by the TIMER2.0 platform. Wilcoxon test was performed, and the *p*-value is indicated in the figure.

**Figure 2 cancers-15-05623-f002:**
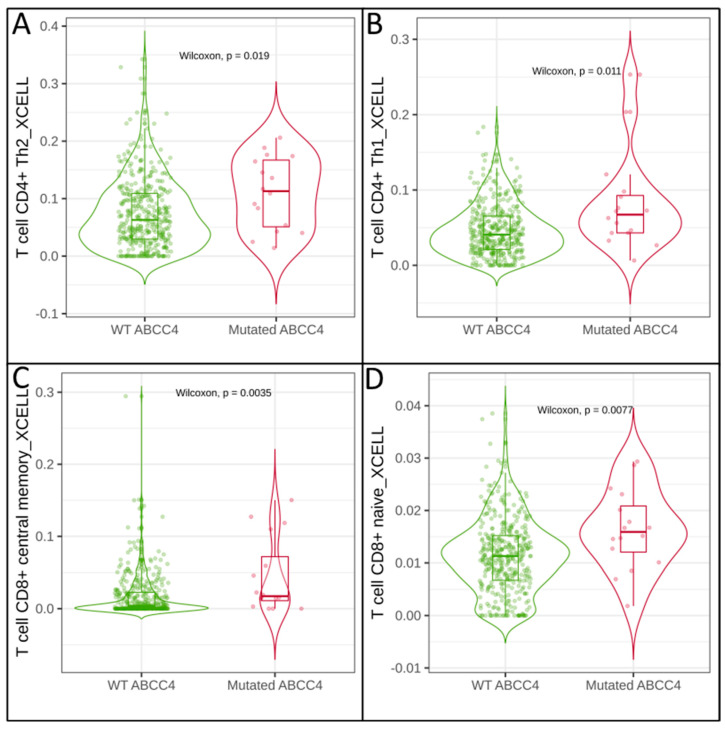
Correlation of mutated *ABCC4* gene expression and immune cell infiltration of CRC. ABCC4 presents a high positive correlation with CD4+ Th2 log2FC = 0.565 (**A**) and CD4+ Th1 log2FC = 0.805 (**B**), CD8+ central memory log2FC = 1.401 (**C**) and CD8+ naive log2FC = 0.526 (**D**) T-cell infiltration. The calculation was performed using TCGA data and visualised by the TIMER2.0 platform. Wilcoxon test was performed, and the *p*-value is indicated in the figure.

**Figure 3 cancers-15-05623-f003:**
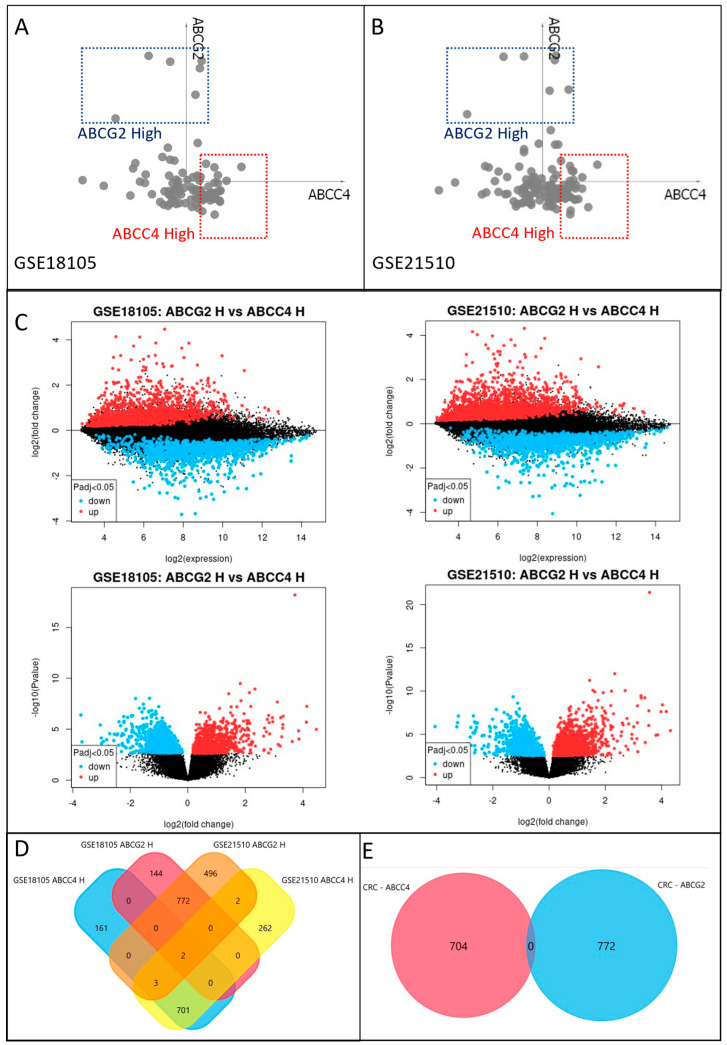
Identification of CRC subgroups presenting high ABCC4 (ABCC4 High) and ABCG2 (ABCG2 High) expression levels and related DEGs. Identification of CRC samples belonging to each subgroup using GSE18105 (**A**) and GSE21510 (**B**) datasets. Visualisation performed using a 2D VizRank-based algorithm and Orange data mining 3.31.1 software. Visual representation of differently expressed genes (DEGs) for ABCG2 High and ABCC4 High CRC subgroups in GSE18105 and GSE21510 analysed using the GEO2R online tool (**C**). Venn diagram of selected ABCG2 High and ABCC4 High DEGs from datasets GSE18105 and GSE21510 (**D**). Venn diagram presenting no mutual DEGs between ABCC4 High and ABCG2 High subgroups (**E**).

**Figure 4 cancers-15-05623-f004:**
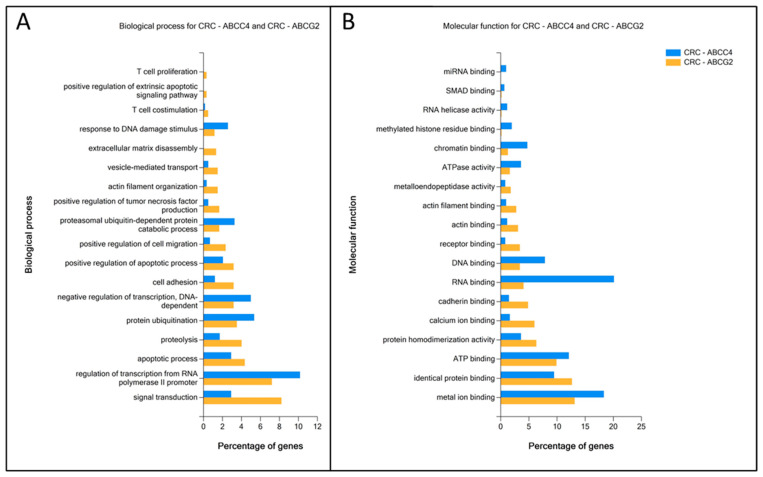
Enrichment analysis of ABCC4 High and ABCG2 High CRC subgroups related to DEGs. Data were analysed and visualised using the FunRich platform supported by the Gene Ontology database, depicting the total percentage of subgroup-specific DEGs enriched in Biological Process (**A**) and Molecular Functions (**B**).

**Figure 5 cancers-15-05623-f005:**
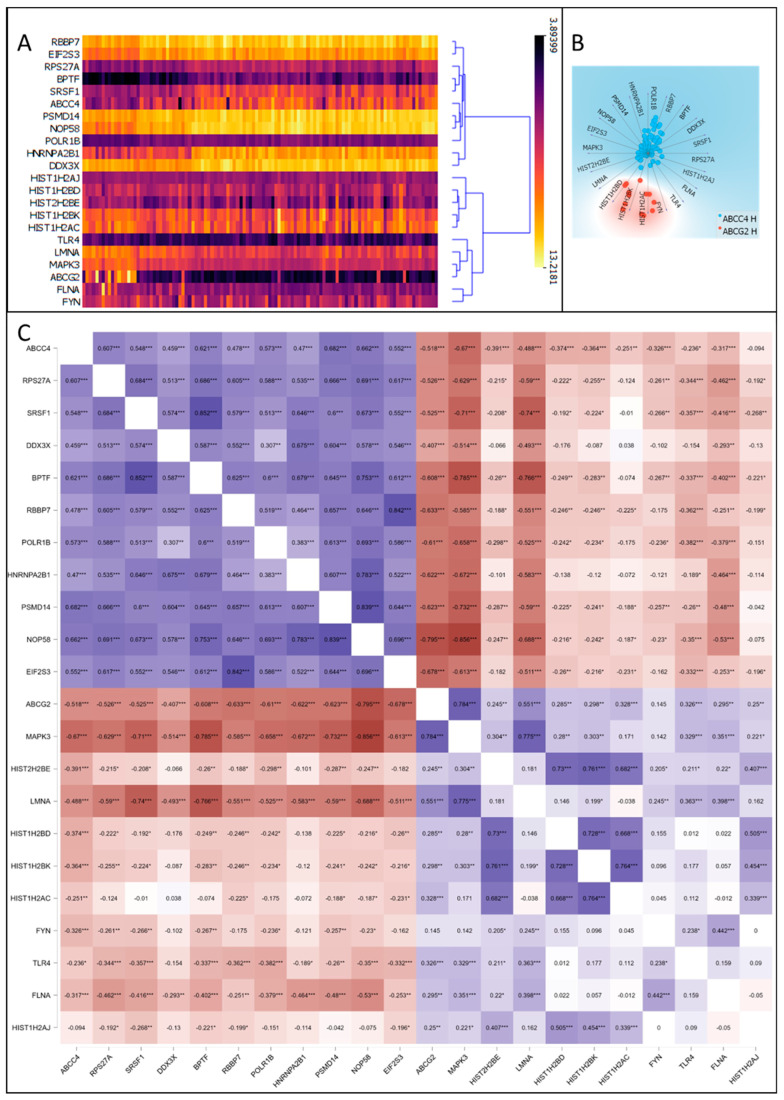
ABCC4 and ABCG2 correlation with top hub proteins selected from ABCC4 High and ABCG2 High CRC subgroups related to DEGs. Hierarchical clustering analysis of chosen top protein hubs, selected from DEGs using mRNA values of CRC patients from GSE18105 and GSE21510 datasets visualised using Orange data mining 3.31.1 software (**A**). Radial visualisation of clusters created by chosen top networking DEGs visualised using the FreeViz tool and Orange data mining 3.31.1 software and mRNA values of CRC patients from GSE18105 and GSE21510 datasets (**B**). Pearson correlation matrix of ABCC4, ABCG2 and chosen DEGs, using mRNA values of CRC patients from GSE18105 and GSE21510 datasets, calculated and visualised using JASP 0.16.0.0 software. * *p* < 0.05; ** *p* < 0.005; *** *p* < 0.001 (**C**).

**Figure 6 cancers-15-05623-f006:**
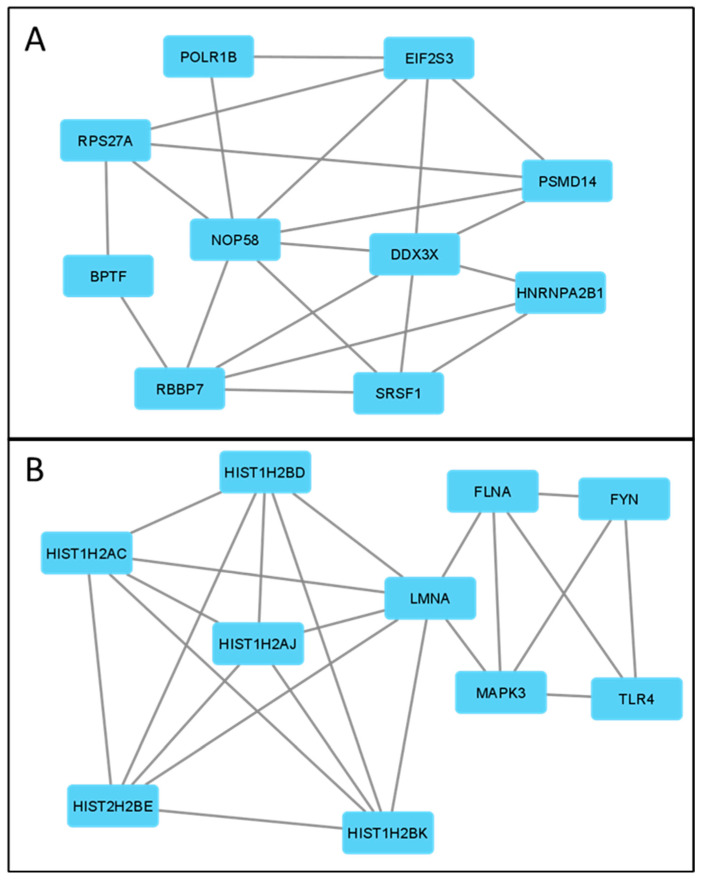
Protein–protein interaction network (PPI) of ABCC4 High (**A**) and ABCG2 High (**B**) CRC subgroups related to top networking DEGs. Calculated using the STRING platform and visualised by Cytoscape 3.9.1.

**Figure 7 cancers-15-05623-f007:**
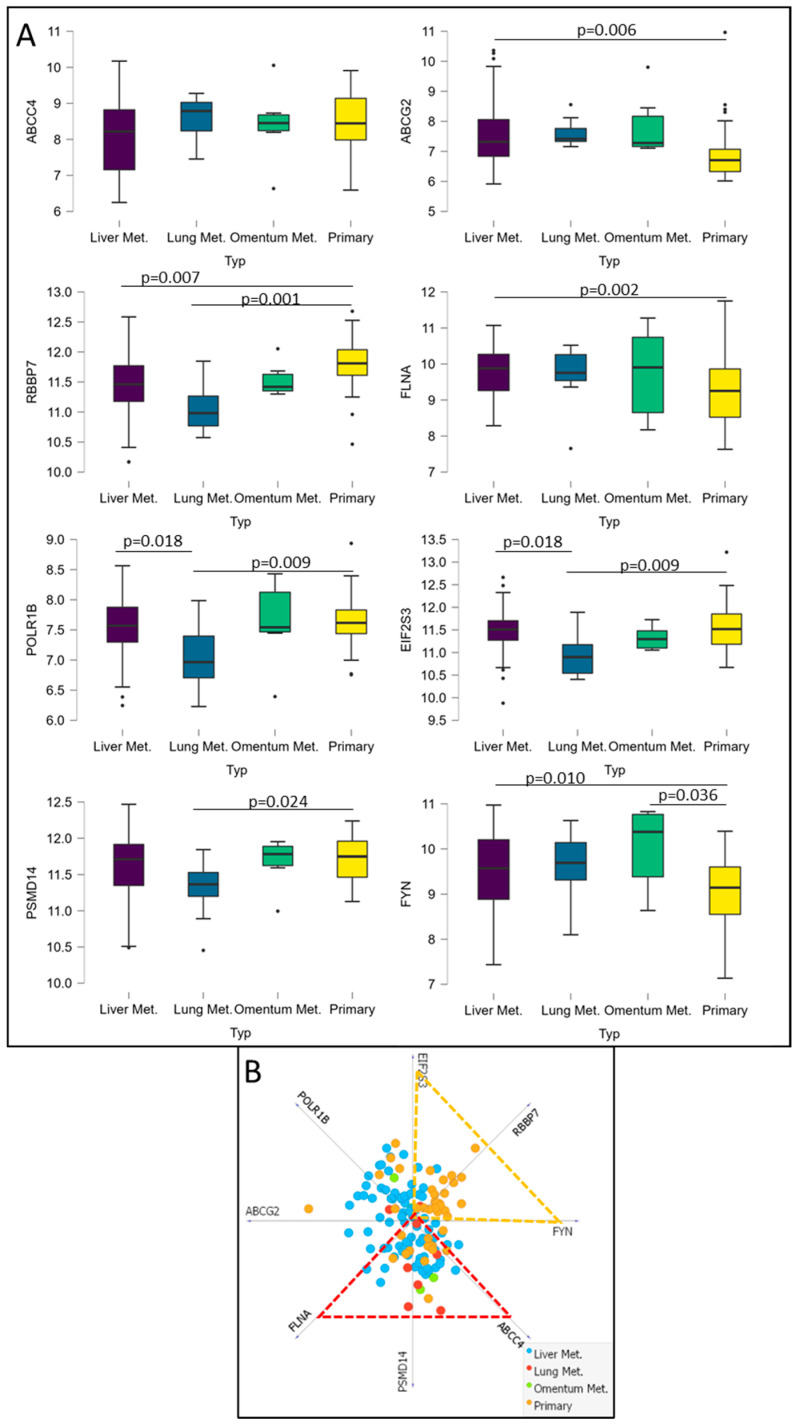
Selected DEGs expression level and its implication as metastatic CRC organotropism biomarkers. mRNA expression levels of selected DEGs in primary CRC, liver, lung and omentum metastases. Data downloaded from GSE41568 were calculated and visualised using JASP 0.16.0.0 software. A normality test (Shapiro–Wilk) was performed, followed by the Mann–Whitney U test (**A**)—linear projection model of primary and metastatic CRC based on the mRNA expression level of chosen DEGs. GSE41568 data were analysed and visualised using Orange data mining 3.31.1 software. Dashed lines presents region of interest formed by chosen DEGs mRNA expression consisting mostly of primary (orange) and metastatic (red) CRC (**B**).

**Figure 8 cancers-15-05623-f008:**
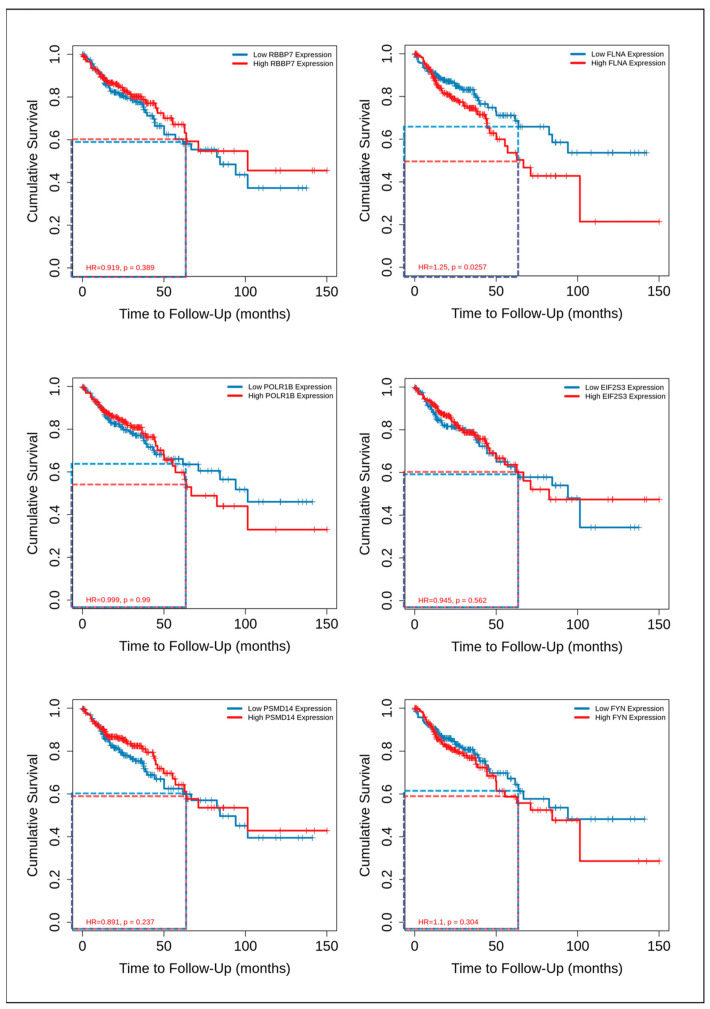
Selected DEGs’ impact on CRC patients’ survival rate. RBBP7, FLNA, POLR1B, EIF2S3, PSMD14 and FYN impact on survival rate was analysed using TCGA data and visualised by the TIMER2.0 platform. Dashed lines indicate 5-year survival rate (60 months) for CRC samples characterised by low (blue) or high (red) expression of chosen DEGs.

**Figure 9 cancers-15-05623-f009:**
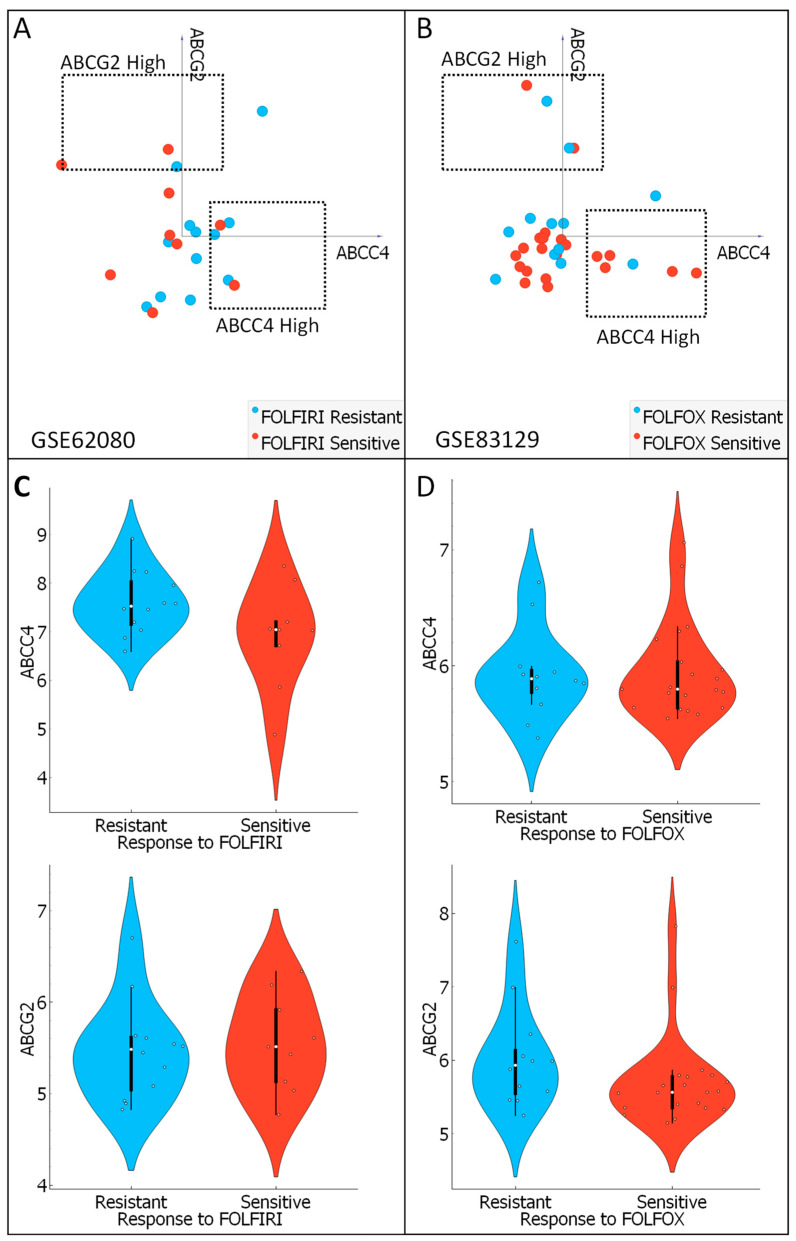
Chemotherapy response predictive capabilities of ABCC4 and ABCG2 mRNA expression. Data containing FOLFIRI (**A**)- and FOLFOX (**B**)-resistant and -sensitive patients’ responses were downloaded from the GEO database (GSE62080 and GSE83129, respectively). Visualisation was performed using a 2D VizRank-based algorithm and Orange data mining 3.31.1 software. Dashed lines present gates set for ABCG2 High and ABCC4 High subgroups. mRNA expression of ABCC4 I ABCG2 in FOLFIRI (**C**)- and FOLFOX (**D**)-treated CRC patients’ sample. Visualisation was performed using Orange data mining 3.31.1 software and the GEO database (GSE62080 and GSE83129, respectively).

**Table 1 cancers-15-05623-t001:** Correlation of immune cell infiltration of CRC tumour subfractions presenting high ABCG2 or ABCC4 expression level. Data were obtained using the TIMER2.0 platform and TCGA database.

	Infiltrating Cells	Correlation Rho	*p*
ABCC4 H			
	CAFs	0.188	1.72 × 10^−3^
	Neutrophils	0.226	8.01 × 10^−6^
	NK	−0.181	2.54 × 10^−3^
	Macrophage	0.326	3.26 × 10^−8^
	Macrophage M1	0.358	9.51 × 10^−10^
	Macrophage M2	0.304	2.86 × 10^−7^
	CD8+ T-cell effector memory	−0.119	4,87 × 10^−2^
	CD8+ T-cell-naive	−0.143	1.75 × 10^−2^
	CD4+ T cell Th1	−0.207	5.41 × 10^−4^
	CD4+ central memory	−0.127	3.50 × 10^−2^
ABCG2 H			
	CAFs	0.182	2.48 × 10^−3^
	Neutrophils	0.119	4.89 × 10^−2^
	Class-switched memory B cells	−0.228	1.35 × 10^−4^
	CD8+ T-cell central memory	−0.14	2.04 × 10^−2^
	CD4+ T-cell effector memory	−0.153	1.12 × 10^−2^
	CD4+ T cell Th2	−0.143	1.75 × 10^−2^
	CD4+ T cell non-regulatory	−0.138	2.20 × 10^−2^

**Table 2 cancers-15-05623-t002:** Main protein hubs observed among DEGs of ABCC4 High and ABCG2 High CRC subgroups.

	Gene	Protein	Number of Edges
ABCC4 H			
	*RPS27A*	ribosomal protein S27a	87
	*SRSF1*	Serine/arginine-rich splicing factor 1	56
	*DDX3X*	DEAD-box helicase family member	50
	*BPTF*	bromodomain PHD finger transcription factor	44
	*RBBP7*	RB Binding Protein 7, Chromatin Remodeling Factor	44
	*POLR1B*	RNA Polymerase I Subunit B	44
	*HNRNPA2B1*	heterogeneous nuclear ribonucleoprotein A2/B1	43
	*PSMD14*	proteasome 26S subunit, non-ATPase 14	42
	*NOP58*	ribonucleoprotein	42
	*EIF2S3*	eukaryotic translation initiation factor 2 subunit gamma	41
ABCG2 H			
	*MAPK3*	mitogen-activated protein kinase 3	53
	*HIST2H2BE*	histone cluster 2 H2B family member E (H2B clustered histone 21)	34
	*LMNA*	Lamin A/C	29
	*HIST1H2BD*	histone cluster 1 H2B family member D (H2B clustered histone 5)	29
	*HIST1H2BK*	histone cluster 1 H2B family member K (H2B clustered histone 12)	29
	*HIST1H2AC*	histone cluster 1 H2A family member C (H2A clustered histone 6)	29
	*FYN*	FYN proto-oncogene, Src family tyrosine kinase	28
	*TLR4*	toll-like receptor 4	28
	*FLNA*	filamin A	28
	*HIST1H2AJ*	histone cluster 1 H2B family member J (H2A clustered histone 14)	27

**Table 3 cancers-15-05623-t003:** Main protein hubs for ABCC4 High and ABCG2 High CRC subgroup KEGG enrichment.

	KEGG ID	Description	Strength	False Discovery Rate
ABCC4 H				
	hsa03050	Proteasome	2.14	2.26 × 10^−9^
	hsa05012	Parkinson’s disease	1.46	5.28 × 10^−7^
	hsa05017	Spinocerebellar ataxia	1.64	5.28 × 10^−7^
	hsa05014	Amyotrophic lateral sclerosis	1.29	3.48 × 10^−6^
	hsa05020	Prion disease	1.35	1.47 × 10^−5^
	hsa05016	Huntington’s disease	1.29	2.41 × 10^−5^
	hsa05010	Alzheimer’s disease	1.22	5.63 × 10^−5^
	hsa05169	Epstein–Barr virus infection	1.31	0.0018
	hsa03020	RNA polymerase	1.8	0.0192
ABCG2 H				
	hsa05034	Alcoholism	1.74	2.90 × 10^−7^
	hsa05133	Pertussis	1.95	4.60 × 10^−7^
	hsa05322	Systemic lupus erythematosus	1.85	9.16 × 10^−7^
	hsa04620	Toll-like receptor signalling pathway	1.81	1.02 × 10^−6^
	hsa05132	Salmonella infection	1.49	2.75 × 10^−5^
	hsa04064	NF-kappa B signalling pathway	1.71	5.70 × 10^−5^
	hsa04217	Necroptosis	1.54	0.00022
	hsa05203	Viral carcinogenesis	1.46	0.00042
	hsa05205	Proteoglycans in cancer	1.43	0.00049
	hsa05235	PD-L1 expression and PD-1 checkpoint pathway in cancer	1.65	0.0014
	hsa05142	Chagas disease	1.6	0.0018
	hsa05145	Toxoplasmosis	1.57	0.0020
	hsa05135	Yersinia infection	1.5	0.0030
	hsa05161	Hepatitis B	1.39	0.0056
	hsa05152	Tuberculosis	1.37	0.0059
	hsa05164	Influenza A	1.38	0.0059
	hsa04621	NOD-like receptor signalling pathway	1.35	0.0060
	hsa05130	Pathogenic Escherichia coli infection	1.32	0.0070
	hsa04510	Focal adhesion	1.3	0.0078
	hsa05131	Shigellosis	1.25	0.0098
	hsa05134	Legionellosis	1.68	0.0137
	hsa04010	MAPK signalling pathway	1.13	0.0185
	hsa04520	Adherens junction	1.59	0.0185
	hsa04664	Fc epsilon RI signalling pathway	1.6	0.0185
	hsa05140	Leishmaniasis	1.57	0.0185
	hsa05221	Acute myeloid leukemia	1.6	0.0185
	hsa05220	Chronic myeloid leukemia	1.54	0.0193
	hsa04012	ErbB signalling pathway	1.5	0.0226
	hsa04660	T-cell-receptor signalling pathway	1.41	0.0313
	hsa05146	Amoebiasis	1.42	0.0313
	hsa04066	HIF-1 signalling pathway	1.39	0.0327
	hsa04725	Cholinergic synapse	1.38	0.0341
	hsa04071	Sphingolipid signalling pathway	1.35	0.0366
	hsa04380	Osteoclast differentiation	1.33	0.0385
	hsa04611	Platelet activation	1.33	0.0385
	hsa04650	Natural-killer-cell-mediated cytotoxicity	1.33	0.0385
	hsa04210	Apoptosis	1.3	0.0418
	hsa04910	Insulin signalling pathway	1.29	0.0418
	hsa04145	Phagosome	1.26	0.0456
	hsa04072	Phospholipase D signalling pathway	1.25	0.0475

## Data Availability

The data presented in this study are available in this article.

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
