# Peer review of "Characteristics of ABCC4 and ABCG2 High Expression Subpopulations in CRC—A New Opportunity to Predict Therapy Response"

_cancers, 2023, doi:10.3390/cancers15235623_

Round 1
Reviewer 1 Report
Comments and Suggestions for Authors
The expression of ABC transporters in various types of cancer and the influence of its level on factors related to the cancer process, as well as on the effectiveness of anticancer therapies, has been a topic of interest in the scientific community for many years. The authors of the manuscript submitted for evaluation, using advanced bioinformatics tools, undertook a highly complex, comprehensive analysis of the associations between high mRNA expression of two transporters ABCC4 and ABCG2 in CRC and molecular and biological features/processes/functions in order to determine their potential use as markers of therapeutic outcome.
I found this manuscript interesting and valuable. The results of the presented study have clinical potential for future use, however, I have some comments for the authors as detailed below.
1. Did the authors compare the mRNA expression profile with the protein expression profile of the studied transporters in CRC on the basis of their study or data in the literature or available databases? The mRNA expression profile may be different from the protein profile.
2. Why did the authors focus on analyzing a small subgroup with high ABCG2 mRNA expression when its expression is mostly downregulated in CRC?
3. DEGs = differently expressed genes. The abbreviation should be defined the first time it appears in the legend of Figure 3 (in the first sentence, not later),
4. DEGs not DEG in the title of the subsection: „3.4. Enrichment analysis of DEG unique to ABCC4…”
5. “ABCC4 High CRC subgroup presents significantly higher enrichment…” should be “The ABCC4 High CRC subgroup…”
6. In the legend of Figure 4: “Enrichment analysis of ABCC2 High and…” should be “ABCC4”. In another sentence: “Biological process” should be “Biological processes”
7. In the first sentence of the legend of Figure 5: “…top hub proteins selected from ABCC2 High…” should be “ABCC4”
8. Remove the underlining and change the font colour in part of the sentence: „The hierarchical clustering method results in a hierarchical dendrogram high-lighting similarities and differences between analysed subjects.” (Materials and methods section, „Hierarchical clustering” subsection),
9. In the first sentence of the second paragraph of the Discussion section: “…CRC patients's sample subfractions…” should be “CRC patients' sample…” or “CRC patient samples”
Author Response
Response to Reviewer #4 Comments
Comment 1. Did the authors compare the mRNA expression profile with the protein expression profile of the studied transporters in CRC based on their study or data in the literature or available databases? The mRNA expression profile may be different from the protein profile.
Response: We thank the reviewer for the valuable comment. In our work, we have analyzed only those databases that contain gene expression profiles since any changes in cell phenotype are first manifested at the gene expression level. These databases collect the most data from patients and are ideal for identifying prognostic or diagnostic factors. Databases containing patient proteomic profiles have much fewer data, and comparing them due to the word limit in the abstract would be highly complicated and challenging and require comparing both profiling methods, i.e., RNA vs. protein analysis. The changes in the expression profile of individual proteins that we have shown are discussed in the Discussion section in the context of the function of the proteins or how they are involved in cellular processes. An analysis of databases containing proteomic profiles is beyond the scope of our manuscript. We believe that this is a topic for future study.
Comment 2. Why did the authors focus on analyzing a small subgroup with high ABCG2 mRNA expression when its expression is mostly downregulated in CRC?
Response: An explanation of why we have focused on the high ABCG2 expression group can be found at the beginning of section 3.3 of the results section. Please refer to the sentence: ,,Even though ABCG2 presents significantly lower expression in CRC samples than in the normal colon, a small subfraction showing a high mRNA level is observed."
Comment 3. DEGs = differently expressed genes. The abbreviation should be defined the first time it appears in the legend of Figure 3 (in the first sentence, not later).
Response: The abbreviation DEGs were defined in the methods section ( in the 2.2 section), where they appear for the first time.
Comment 4. DEGs not DEG in the title of the subsection: "3.4. Enrichment analysis of DEG unique to ABCC4…"
Response: The title of the subsection was corrected.
Comment 5. “ABCC4 High CRC subgroup presents significantly higher enrichment…” should be “The ABCC4 High CRC subgroup…”
Response: The sentence was corrected.
Comment 6. In the legend of Figure 4: "Enrichment analysis of ABCC2 High and…" should be "ABCC4". In another sentence: "Biological process" should be "Biological processes"
Response: The sentences were corrected.
Comment 7. In the first sentence of the legend of Figure 5: "…top hub proteins selected from ABCC2 High…" should be "ABCC4"
Response: The sentence was corrected.
Comment 8. Remove the underlining and change the font colour in part of the sentence: "The hierarchical clustering method results in a hierarchical dendrogram highlighting similarities and differences between analyzed subjects." (Materials and methods section, "Hierarchical clustering" subsection),
Response: The underlining and additional font colour were editorial additions.
Comment 9. In the first sentence of the second paragraph of the Discussion section: "…CRC patients' sample subfractions…" should be "CRC patients' sample…" or "CRC patient samples"
Response: The sentence was corrected.
Reviewer 2 Report
Comments and Suggestions for Authors
The authors provide a study of clinical value of ABC transporters ABCC4 and ABCG2 based on gene expression data of three public datasets of CRC. The analysis and choice of two candidate genes is based on their previously published results. This study presents some new interesting findings and revelations in these highly studied genes. Besides others, they found that ABCC4 and ABCG2 levels present a negative correlation with CD4+ and CD8+ T cell infiltration and positive correlation with CAF infiltration. In addition, they divided the datasets into two subpopulations based on high ABCC4 and ABCG2 levels. Finally, they identified protein clusters associated with apoptosis and EMT processes. Transcript levels of ABCG2 and of genes from seven hub proteins also predicted CRC metastatic potential and site (e.g. deregulation of ABCG2 and some other genes characterized liver metastases). The findings are interesting; however, the manuscript lacks a clear message about potential clinical or research application. Also the potential to predict therapy is not substantiated enough.
Major comments:
1/ Authors claim that patients can be separated based on gene expression of ABCC4 and ABCG2 genes into subgroups with different susceptibility to therapy response. However, without an additional dataset with gene expression levels and known therapy response, it is not possible to verify such claim. If no validation dataset can be provided, the statements should be toned down in Title, Abstract and Conclusion.
2/ Please add brief summary about the clinical meaning applicability of your findings in the Abstract and Discussion.
Some minor points:
1/ Abstract and page 18. Please correct: Ant-EGFR therapy.
2/ Page 4. Mutation analysis of ABC4/G2. Please specify what does “mutated variant of ABCC4...” mean? Do you mean some special genetic variant or is it just any mutation/s in the respective genes?
3/ Figure 1H. Please correct the x-axis labels (should be ABCG2?)
4/ Page 6, second line of section 3.2. Please add the word “fibroblasts” in: …cancer-associated (CAFs)
5/ Figure 3A, B. Please provide labels for x and y axes.
6/ Page 9, Line 7 of 3.4. Missing comma between “cell adhesion” and “extracellular matrix”.
7/ Figure 5. Please provide this figure in higher resolution.
8/ Page 13. Please provide the number of patients for dataset GSE41568.
9/ Page 14 and Figure 7. Authors describe the potential of several genes to predict metastatic site. E.g.: “Upregulation of ABCG2, FLNA and FYN with simultaneous downregulation of RBBP7 is characteristic of CRC metastasis to the liver… ABCG2 High CRC subgroup prefers liver metastasis. Additionally, POLR1B and EIF2S3 expression downregulation… was observed for CRC samples resected from the lung.” The presence of distant metastasis is a negative prognostic factor as mCRC has a very poor prognosis, especially mCRC with synchronous metastases. It would be good to know if any of the mentioned genes have prognostic value. The authors should present survival analysis from suitable dataset, e.g. GEO or TCGA.
10/ Page 14. Is it possible to predict the risk (of the presence) of metastasis based on the presented model? Please add a short discussion.
11/ Discussion. The authors should declare the limitations of the study. For example, what is the ethnicity of patients from the used datasets and how does it affect the study reliability in different populations?
Author Response
Comment 1: Authors claim that patients can be separated based on gene expression of ABCC4 and ABCG2 genes into subgroups with different susceptibility to therapy response. However, without an additional dataset with gene expression levels and known therapy response, it is not possible to verify such claim. If no validation dataset can be provided, the statements should be toned down in Title, Abstract and Conclusion.
Response: We thank the reviewer for this valuable suggestion. We added the requested analysis of additional data sheets. The results of this analysis are presented in Figure 9, appropriately described in the results section, and discussed in the discussion section.
Comment 2: Please add brief summary about the clinical meaning applicability of your findings in the Abstract and Discussion.
Response: Due to the word limit in the abstract section, a brief summary of the clinical significance of our analysis has been added at the end of the discussion and conclusion section as well as to the graphical abstract.
Some minor points:
Comment 1: Abstract and page 18. Please correct: Ant-EGFR therapy.
Response: The phrase was corrected.
Comment 2: Page 4. Mutation analysis of ABC4/G2. Please specify what does "mutated variant of ABCC4..." mean? Do you mean some special genetic variant or is it just any mutation/s in the respective genes?
Response: We were referring to any mutation in the ABCC4 gene. An appropriate explanation was added in the results section.
Comment 3: Figure 1H. Please correct the x-axis labels (should be ABCG2?)
Response: The x-axis labels are accurate.
Comment 4: Page 6, second line of section 3.2. Please add the word "fibroblasts" in: …cancer-associated (CAFs)
Response: The word ,, fibroblasts" was added.
Comment 5: Figure 3A, B. Please provide labels for x and y axes.
Response: The x-axis labels are accurate. To clarify the interpretation of this figure, we have rearranged Figure 3A, B.
Comment 6: Page 9, Line 7 of 3.4. Missing comma between "cell adhesion" and "extracellular matrix".
Response: The missing comma was added.
Comment 7: Figure 5. Please provide this figure in higher resolution.
Response: We replaced the figure, and we have provided the figure in as high a resolution as possible.
Comment 8: Page 13. Please provide the number of patients for dataset GSE41568.
Response: The number of patients was added in the results section.
Comment 9: Page 14 and Figure 7. Authors describe the potential of several genes to predict metastatic sites. e.g.: "Upregulation of ABCG2, FLNA and FYN with simultaneous downregulation of RBBP7 is characteristic of CRC metastasis to the liver… ABCG2 High CRC subgroup prefers liver metastasis. Additionally, POLR1B and EIF2S3 expression downregulation… was observed for CRC samples resected from the lung." The presence of distant metastasis is a negative prognostic factor as mCRC has a very poor prognosis, especially mCRC with synchronous metastases. It would be good to know if any of the mentioned genes have prognostic value. The authors should present survival analysis from suitable dataset, e.g. GEO or TCGA.
Response: We thank the reviewer for this valuable suggestion. We added the survival analysis in Figure 8 and discussed it in the revised version of the manuscript.
Comment 10: Page 14. Is it possible to predict the risk (of the presence) of metastasis based on the presented model? Please add a short discussion.
Response: We added a short paragraph in the discussion section.
Comment 11: Discussion. The authors should declare the limitations of the study. For example, what is the ethnicity of patients from the used datasets and how does it affect the study reliability in different populations?
Response: We added a short paragraph in the conclusion section.
Reviewer 3 Report
Comments and Suggestions for Authors
The manuscript was aimed to define the prognostic value of highly expressed ABCC4 and ABCG2 in patients with colorectal cancer. The authors concluded that population of CRC patients with highly expressed ABCC4 exhibited EMT reprogramming and high response to DNA damage stimuli. In contrast, patients with CRC exhibited high expression of ABCG2 might be resistant ot anti-EGFR-therapies.
In general, the manuscript is very well prepared and raising up an important scientific topic about the prognostic values of these ABC-transporters.
Minor:
1) Given that the certain chemotherapeutic agents used for cancer therapy exhibit potent DNA-damaging activities and are also known to be the substrates for ABC-transporters, it will be important to explain the data illustrating the relationship between high expression of ABCC4 and high response to DNA damage stimuli. I guess that "response to DNA damage stimuli" shown in the manuscript means the activation of DNA repair. If so, activation of DNA repair might be result of the extensive DNA damage induced due to retention of the chemotherapeutic agents from cancer cells. In this case, tumors with high ABCC4 expression should exhibit low response to DNA damage stimuli due to the ABC-mediated efflux of DNA damaging agents.
2) Discussion section contains some duplications - e.g. "Approximately 40-50% of CRC presents resistance to antiepidermal growth factor receptor (anti-EGFR) therapy"
Author Response
Comment 1: Given that the certain chemotherapeutic agents used for cancer therapy exhibit potent DNA-damaging activities and are also known to be the substrates for ABC-transporters, it will be important to explain the data illustrating the relationship between high expression of ABCC4 and high response to DNA damage stimuli. I guess that "response to DNA damage stimuli" shown in the manuscript means the activation of DNA repair. If so, activation of DNA repair might be result of the extensive DNA damage induced due to retention of the chemotherapeutic agents from cancer cells. In this case, tumors with high ABCC4 expression should exhibit low response to DNA damage stimuli due to the ABC-mediated efflux of DNA damaging agents.
Response: We thank the reviewer for this valuable comment. In our manuscript, we have demonstrated that ,,the ABCC4 High CRC subset shows a significantly higher enrichment in processes related to DNA and RNA binding, regulation of gene expression and response to DNA damage”. The analyses we have carried out are based on strictly theoretical simulations, and the generation of a specific hypothesis, as indicated above, would need to be verified under experimental conditions. By demonstrating the association of high ABCC4 expression with different biological processes in the tumour cell, we would rather like to point out the physiological role of the ABCC4 protein, e.g. as an exporter of signalling molecules during the process of tumour progression, to link it to stages of metastasis development and to ask whether it could be prognostic factor rather than as a transporter of a chemotherapeutic agent.
Comment 2: Discussion section contains some duplications - e.g. "Approximately 40-50% of CRC presents resistance to antiepidermal growth factor receptor (anti-EGFR) therapy"
Response: The discussion section was corrected, and the duplications were removed.
Reviewer 4 Report
Comments and Suggestions for Authors
Comments to the article cancers-2717644. This is an interesting and well-written article in which the authors evaluate the expression of the ABCC4 and ABCG2 transporters in subpopulations in patients with colorectal cancer to determine their potential use as markers of response to chemotherapy.
Just a small recommendation is to provide the electronic addresses of the databases used and the access data
Author Response
Comment 1: Just a small recommendation is to provide the electronic addresses of the databases used and the access data,
Response: The electronic addresses of the databases and access data were added.
Round 2
Reviewer 2 Report
Comments and Suggestions for Authors
Authors answered all comments. I recommend accept in present form.